# Emergent Road Rules In Multi-Agent Driving Environments

**Avik Pal[1], Jonah Philion[2,3,4], Yuan-Hong Liao[2,4], Sanja Fidler[2,3,4]**
[1]IIT Kanpur, [2]University of Toronto, [3]NVIDIA, [4]Vector Institute
`avikpal@cse.iitk.ac.in`, `{jphilion, andrew, fidler}@cs.toronto.edu`

## Abstract

For autonomous vehicles to safely share the road with human drivers, autonomous vehicles must abide by specific "road rules" that human drivers have agreed to follow. "Road rules" include rules that drivers are required to follow by law – such as the requirement that vehicles stop at red lights – as well as more subtle social rules – such as the implicit designation of fast lanes on the highway. In this paper, we provide empirical evidence that suggests that – instead of hard-coding road rules into self-driving algorithms – a scalable alternative may be to design multi-agent environments in which road rules emerge as optimal solutions to the problem of maximizing traffic flow. We analyze what ingredients in driving environments cause the emergence of these road rules and find that two crucial factors are noisy perception and agents' spatial density. We provide qualitative and quantitative evidence of the emergence of seven social driving behaviors, ranging from obeying traffic signals to following lanes, all of which emerge from training agents to drive quickly to destinations without colliding. Our results add empirical support for the social road rules that countries worldwide have agreed on for safe, efficient driving.

## 1 Introduction

Public roads are significantly more safe and efficient when equipped with conventions restricting how one may use the roads. These conventions are, to some extent, arbitrary. For instance, a "drive on the left side of the road" convention is, practically speaking, no better or worse than a "drive on the right side of the road" convention. However, the decision to reserve *some* orientation as the canonical orientation for driving is far from arbitrary in that establishing such a convention improves both safety (doing so decreases the probability of head-on collisions) and efficiency (cars can drive faster without worrying about dodging oncoming traffic).

**Figure 1. Multi-agent Driving Environment** We train agents to travel from a→b as quickly as possible with limited perception while avoiding collisions and find that "road rules" such as lane following and traffic light usage emerge.

In this paper, we investigate the extent to which these road rules – like the choice of a canonical heading orientation – can be learned in multi-agent driving environments in which agents are trained to drive to different destinations as quickly as possible without colliding with other agents. As visualized in Figure 1, our agents are initialized in random positions in different maps (either synthetically generated or scraped from real-world intersections from the nuScenes dataset (Caesar et al., 2019)) and tasked with reaching a randomly sampled feasible target destination. Intuitively, when agents have full access to the map and exact states of other agents, optimizing for traffic flow leads the agents to drive in qualitatively aggressive and un-humanlike ways. However, when perception is imperfect and noisy, we show in Section 5 that the agents begin to rely on constructs such as lanes, traffic lights, and safety distance to drive safely at high speeds.

Notably, while prior work has primarily focused on building driving simulators with *realistic sensors* that mimic LiDARs and cameras (Dosovitskiy et al., 2017; Manivasagam et al., 2020; Yang et al., 2020; Bewley et al., 2018), we focus on the high-level design choices for the simulator – such as the definition of reward and perception noise – that determine if agents trained in the simulator exhibit *realistic behaviors*. We hope that the lessons in state space, action space, and reward design gleaned

from this paper will transfer to simulators in which the prototypes for perception and interaction used in this paper are replaced with more sophisticated sensor simulation. Code and Documentation for all experiments presented in this paper can be found in our Project Page[1].

Our main contributions are as follows:

- We define a multi-agent driving environment in which agents equipped with noisy LiDAR sensors are rewarded for reaching a given destination as quickly as possible without colliding with other agents and show that agents trained in this environment learn road rules that mimic road rules common in human driving systems.
- We analyze what choices in the definition of the MDP lead to the emergence of these road rules and find that the most important factors are perception noise and the spatial density of agents in the driving environment.
- We release a suite of 2D driving environments[2] with the intention of stimulating interest within the MARL community to solve fundamental self-driving problems.

## 2 RELATED WORKS

**Reinforcement Learning** Deep Reinforcement Learning (DeepRL) has become an popular framework that has been successfully used to solve Atari (Mnih et al., 2013), Strategy Games (Peng et al., 2017; OpenAI, 2018), and Traffic Control (Wu et al., 2017a; Belletti et al., 2018). Vanilla Policy Gradient (Sutton et al., 2000) is an algorithm that optimizes an agent's policy by using monte-carlo estimates of the expected return. Proximal Policy Optimization (Schulman et al., 2017) – which we use in this work – is an on-policy policy gradient algorithm that alternately samples from the environment and optimizes the policy using stochastic gradient descent. PPO stabilizes the Actor's training by limiting the step size of the policy update using a clipped surrogate objective function.

**Multi-Agent Reinforcement Learning** The central difficulties of Multi-Agent RL (MARL) include environment non-stationarity (Hernandez-Leal et al., 2019; 2017; Busoniu et al., 2008; Shoham et al., 2007), credit assignment (Agogino and Tumer, 2004; Wolpert and Tumer, 2002), and the curse of dimensionality (Busoniu et al., 2008; Shoham et al., 2007). Recent works (Son et al., 2019; Rashid et al., 2018) have attempted to solve these issues in a centralized training decentralized execution framework by factorizing the joint action-value Q function into individual Q functions for each agent. Alternatively, MADDPG (Lowe et al., 2017) and PPO with Centralized Critic (Baker et al., 2019) have also shown promising results in dealing with MARL Problems using policy gradients.

**Emergent Behavior** Emergence of behavior that appears human-like in MARL (Leibo et al., 2019) has been studied extensively for problems like effective tool usage (Baker et al., 2019), ball passing and interception in 3D soccer environments (Liu et al., 2019), capture the flag (Jaderberg et al., 2019), hide and seek (Chen et al., 2020; Baker et al., 2019), communication (Foerster et al., 2016; Sukhbaatar et al., 2016), and role assignment (Wang et al., 2020). For autonomous driving and traffic control (Sykora et al., 2020), emergent behavior has primarily been studied in the context of imitation learning (Bojarski et al., 2016; Zeng et al., 2019; Bansal et al., 2018; Philion and Fidler, 2020; Bhattacharyya et al., 2019). Zhou et al. (2020) solve a similar problem as ours from the perspective of environment design but fail to account for real-world aspects like noisy perception, which are inherent for emergent rules. In contrast to works that study emergent behavior in mixed-traffic autonomy (Wu et al., 2017b), embedded rules in reward functions (Medvet et al., 2017; Talamini et al., 2020) and traffic signal control (Stevens and Yeh, 2016), we consider a fully autonomous driving problem in a decentralized execution framework and show the emergence of standard traffic rules that are present in transportation infrastructure.

## 3 PROBLEM SETTING

We frame the task of driving as a discrete time Multi-Agent Dec-POMDP (Oliehoek et al., 2016). Formally, a Dec-POMDP is a tuple $G = \langle \mathcal{S}, \mathcal{A}, \mathcal{P}, r, \rho_0, \mathcal{O}, n, \gamma, T \rangle$. $\mathcal{S}$ denotes the state space of the environment, $\mathcal{A}$ denotes the joint action space of the $n$ agents s.t. $\bigcup_{i=1}^{n} a_i \in \mathcal{A}$, $\mathcal{P}$ is the state transition probability $\mathcal{P} : \mathcal{S} \times \mathcal{A} \times \mathcal{S} \mapsto \mathbb{R}_+$, $r$ is a bounded reward function $r : \mathcal{S} \times a \mapsto \mathbb{R}$, $\rho_0$ is the initial state distribution, $O$ is the joint observation space of the $n$ agents s.t. $\bigcup_{i=1}^{n} o_i \in \mathcal{O}$, $\gamma \in (0, 1]$ is the discount factor, and $T$ is the time horizon.

---

[1] http://fidler-lab.github.io/social-driving
[2] https://github.com/fidler-lab/social-driving

We parameterize the policy $\pi_\theta : o \times a \mapsto \mathbb{R}_+$ of the agents using a neural network with parameters $\theta$. In all our experiments, the agents share a common policy network. Let the expected return for the $i^{th}$ agent be $\eta_i(\pi_\theta) = \mathbb{E}_\tau \left[ \sum_{t=0}^{T-1} \gamma^t r_{i,t}(s_t, a_{i,t}) \right]$, where $\tau = (s_0, a_{i,0}, \ldots, s_{T-1}, a_{i,T-1})$ is the trajectory of the agent, $s_0 \sim \rho_0$, $a_{i,t} \sim \pi_\theta(a_{i,t}|o_{i,t})$, and $s_{t+1} \sim \mathcal{P}(s_{t+1}|s_t, \bigcup_{i=1}^n a_{i,t})$. Our objective is to find the optimal policy which maximizes the utilitarian objective $\sum \eta_i$.

**Reward** We use high-level generic rewards and avoid any extensive reward engineering. The agents receive a reward of +1 if they successfully reach their given destination and -1 if they collide with any other agent or go off the road. In the event of a collision, the simulation for that agent halts. In an inter-agent collision, we penalize both agents equally without attempting to determine which agent was responsible for the collision. We additionally regularize the actions of the agents to encourage smooth actions and add a normalized penalty proportional to the longitudinal distance of the agent from the destination to encourage speed. We ensure that the un-discounted sum of each component of the reward for an agent over the entire trajectory is bounded $0 \leq r \leq 1$. We combine all the components to model the reward $r_{i,t}$ received by agent $i$ at timestep $t$ as follows:

$$
\begin{aligned}
r_{i,t}(s_t, a_{i,t}) = & \ \mathbb{I}\left[\text{Reached the Destination at timestep } t\right] \\
& - \mathbb{I}\left[\text{Collided for the first time with an obstacle/agent at timesetep } t\right] \\
& - \frac{1}{T} \cdot \left\| \frac{a_{i,t} - a_{i,(t-1)}}{(a_i)_{max}} \right\|_2 - \frac{1}{T} \cdot \frac{\text{Distance to goal from current position}}{\text{Distance to goal from initial position}}
\end{aligned}
$$

where $(a_i)_{max}$ is the maximum magnitude of action that agent $i$ can take

**Map and Goal Representation** We use multiple environments: four-way intersection, highway tracks, and real-world road patches from nuScenes (Caesar et al., 2019), to train the agents. The initial state distribution $\rho_0$ is defined by the drivable area in the base environment. The agents "sense" a traffic signal if they are near a traffic signal and facing the traffic signal. These signals are represented as discrete values – 0, 0.5, 1.0 for the 3 signals and 0.75 for no signal available – in the observation space. In all but our communication experiments, agents have the ability to communicate exclusively through the motion that other agents observe. In our communication experiments, we open a discrete communication channel designed to mimic turn signals and discuss the direct impact on agent behavior. Additionally, to mimic a satnav dashboard, the agents observe the distance from their goal, the angular deviation of the goal from the current heading, and the current speed.

**LiDAR observations** We simulate a LiDAR sensor for each agent by calculating the distance to the closest object – dynamic or static – along each of $n$ equi-angular rays. We restrict the range of the LiDAR to be 50m. Human eyes themselves are imperfect sensors and are easily thwarted by weather, glare, or visual distractions; in our experiments, we study the importance of this "visual" sensor by introducing noise in the sensor. We introduce $x\%$ lidar noise by random uniformly dropping (assigning a value of 0) $x\%$ of the rays at every timestep (Manivasagam et al., 2020). To give agents the capacity to infer the velocity and acceleration of nearby vehicles, we concatenate the LiDAR observations from $T = 5$ past timesteps.

## 4 POLICY OPTIMIZATION

### 4.1 POLICY PARAMETERIZATION

In our experiments we consider the following two parameterizations for our policy network(s):

1. **Fixed Track Model**: We optimize policies that output a Multinomial Distribution over a fixed set of discretized acceleration values. This distribution is defined by $\pi_\phi(\mathfrak{a}|\mathfrak{o})$, where $\pi_\phi$ is our policy network, $\mathfrak{a}$ is the acceleration, and $\mathfrak{o}$ is the observation. This acceleration is used to drive the vehicle along a fixed trajectory from the source to target destination. This model trains efficiently but precludes the emergence of lanes.

2. **Spline Model**: To train agents that are capable of discovering lanes, we use a two-stage formulation inspired by Zeng et al. (2019) in which trajectories shapes are represented by clothoids and time-dependence is represented by a velocity profile. Our overall policy is factored into two "subpolicies" – a spline subpolicy and an acceleration subpolicy. The spline subpolicy is tasked with predicting the spline along which the vehicle is supposed to be driven. This subpolicy conditions on an initial local observation of the environment and

predicts the spline (see Section 5). We use a Centripetal Catmull Rom Spline (Catmull and Rom, 1974) to parameterize this spatial path. The acceleration subpolicy follows the same parameterization from Fixed Track Model, and controls the agent's motion along this spline. We formalize the training algorithm for this bilevel problem in Section 4.4.

Note that the fixed track model is a special case of the spline model parameterization, where the spline is hard-coded into the environment. These splines can be extracted from lane graphs such as those found in the HD maps provided by the nuScenes dataset (Caesar et al., 2019).

## 4.2 PROXIMAL POLICY OPTIMIZATION USING CENTRALIZED CRITIC

We consider a Centralized Training with Decentralized Execution approach in our experiments. During training, the critic has access to all the agents' observations while the actors only see the local observations. We use Proximal Policy Optimization (PPO) (Schulman et al., 2017) and Generalized Advantage Estimation (GAE) (Schulman et al., 2015) to train our agents. Let $V_\phi$ denote the value function. To train our agent, we optimize the following objective:

$$L_1(\phi) = \mathbb{E}\big[\min(\tilde{r}(\phi)\hat{A}, \text{ clip}(\tilde{r}(\phi), 1 - \epsilon, 1 + \epsilon)\hat{A}) - c_1(V_\phi(s, a) - V_{target}) - c_2 H(s, \pi_\phi)\big]$$

where $\tilde{r}(\phi) = \frac{\pi_\phi(a|o)}{\pi_{\phi_{old}}(a|o)}$, $\hat{A}$ is the Estimated Advantage, $H(\cdot)$ measures entropy, $\{\epsilon, c_1, c_2\}$ are hyperparameters, and $V_{target}$ is the value estimate recorded during simulation. Training is performed using a custom adaptation of SpinningUp (Achiam, 2018) for MARL and Horovod (Sergeev and Balso, 2018). The agents share a common policy network in all the reported experiments. In our environments, the number of agents present can vary over time, as vehicles reach their destinations and new agents spawn. To enforce permutation invariance across the dynamic pool of agents, the centralized critic takes as input the mean of the latent vector obtained from all the observations.

---

**Algorithm 1:** Alternating Optimization for Spline and Acceleration Control

**Result:** Trained Subpolicies $\pi_\theta$ and $\pi_\phi$
$\pi_\theta \leftarrow$ Spline Subpolicy, $\pi_\phi \leftarrow$ Acceleration Subpolicy, $V_\phi \leftarrow$ Value Function for Acceleration Control;
**for** $i = 1 \ldots N$ **do**
    /* Given $\pi_\phi$ optimize $\pi_\theta$                                                */
    **for** $k = 1 \ldots K_1$ **do**
        Collect set of Partial Trajectories $\mathcal{D}_{1,k}$ using $a_s \sim \pi_\theta(o_s)$ and $a_a \leftarrow \arg\max \pi_\phi(a|o_a)$;
        Compute and store the normalized rewards $\overline{R}$;
    **end**
    Optimize the parameters $\theta$ using the objective $L_2(\theta)$ and stored trajectories $\mathcal{D}_1$;
    /* Given $\pi_\theta$ optimize $\pi_\phi$ and $vf_\phi$                            */
    **for** $k = 1 \ldots K_2$ **do**
        Collect set of Partial Trajectories $\mathcal{D}_{2,k}$ using $a_s \leftarrow \arg\max \pi_\theta(a|o_s)$ and $a_a \sim \pi_\phi(o_a)$;
        Compute and store the advantage estimates $\hat{A}$ using GAE;
    **end**
    Optimize the parameters $\phi$ using the objective $L_2(\phi)$ and stored trajectories $\mathcal{D}_2$;
**end**

---

## 4.3 SINGLE-STEP PROXIMAL POLICY OPTIMIZATION

In a single-step MDP, the expected return modelled by the critic is equal to the reward from the environment as there are no future timesteps. Hence, optimizing the critic is unnecessary in this context. Let $\overline{R}_t$ denote the normalized reward. The objective function defined in Sec 4.2 reduces to:

$$L_2(\theta) = \mathbb{E}\left[\min(\tilde{r}(\theta)\overline{R}, \text{clip}(\tilde{r}(\theta), 1 - \epsilon, 1 + \epsilon)\overline{R}) - c_2 H(s, \pi_\theta)\right] \quad (1)$$

## 4.4 BILEVEL OPTIMIZATION FOR JOINT TRAINING OF SPLINE/ACCELERATION SUBPOLICIES

In this section, we present the algorithm we use to jointly train two RL subpolicies where one subpolicy operates in a single step and the other operates over a time horizon $T \geq 1$. The subpolicies operate oblivious of each other, and cannot interact directly. The reward for the spline subpolicy is the undiscounted sum of the rewards received by the acceleration subpolicy over the time horizon. Pseudocode is provided in Algorithm 1. The algorithm runs for a total of $N$ iterations. For each

iteration, we collect $K_1$ and $K_2$ samples from the environment to train the spline and acceleration subpolicy respectively. We denote actions and observations using $(a_s, o_s)$ and $(a_a, o_a)$ (not to be confused with the notation used in Section 3) for the spline and acceleration subpolicy respectively.

## 5 EMERGENT SOCIAL DRIVING BEHAVIOR

We describe the various social driving behaviors that emerge from our experiments and analyze them quantitatively. Experiments that do not require an explicit emergence of lanes – 5.1, 5.3, 5.4, 5.7 – use the fixed track model. We use the Spline Model for modeling lane emergence in Experiments 5.2 and 5.5. The results in Section 5.6 are a consequence of the increased number of training agents, and are observed using either agent type. Qualitative rollouts are provided in our project page.

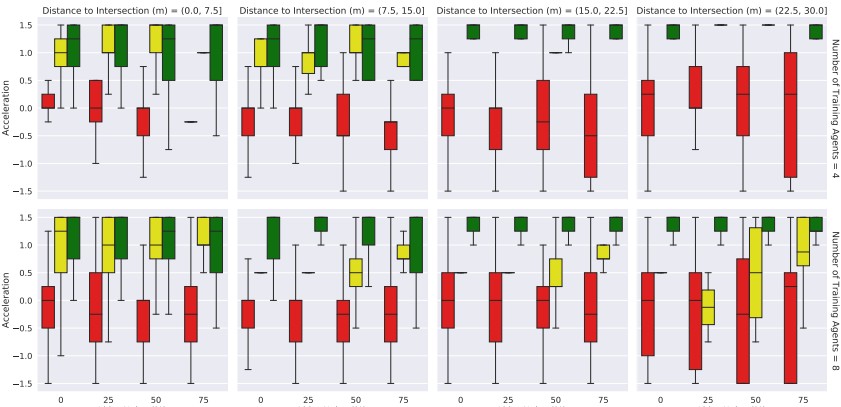

**Figure 2. Traffic Light Usage** Actions taken by the agents with varying amount of perception noise and traffic signal (represented by color coding the box plots). A strong correlation exists between the acceleration, traffic signal and distance from the intersection. As the agent approaches the intersection, the effect of red signal on the actions is more prominent (characterized by the reduced variance). However, the variance for green signal increasing, since agents need to marginally slow down upon detecting an agent in front of them. With more LiDAR noise, detecting the location of the intersection becomes diffcult so the agents prematurely slow down far from the intersection. Agents with better visibility can potentially safely cross the intersection on a red light, hence the mean acceleration near the intersection goes down with increasing lidar noise. Finally, with increasing number of agents, the variability of the actions increase due to presence of leading vehicles on the same path.

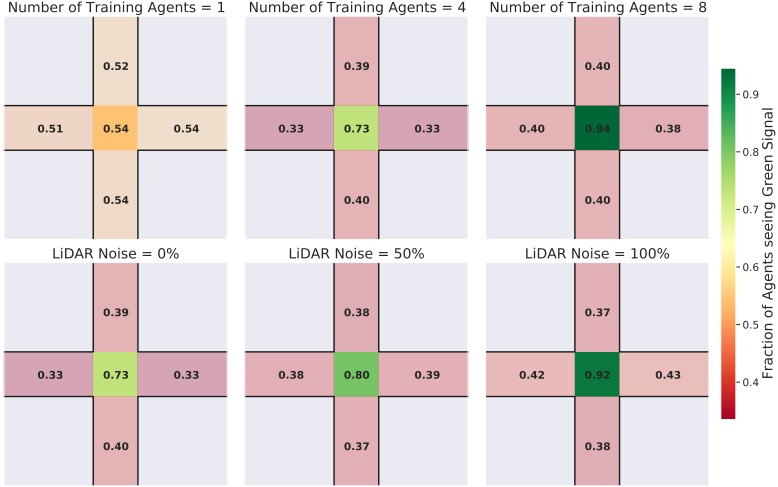

**Figure 3. Traffic light usage** Spatial 2D histogram of a synthetic intersection showing the fraction of agents that see a green signal. Once agents have entered the intersection we consider the signal they saw just before entering it. A darker shade of green in the intersection shows that fewer agents violated the traffic signal. In a single agent environment, there is no need to follow the signals. Agents trained on an environment with higher spatial density (top row) violate the signal less frequently. Agents also obey the signals more with increased perception noise (bottom row).

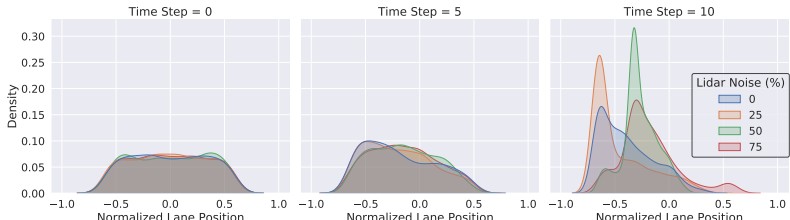

**Figure 4. Lanes emerge with more perception noise** When perception noise is increased agents follow lanes more consistently (higher peaks for 25% and 50% LiDAR models). However, after a certain threshold imperfect perception leads a poor convergence as can be seen for the 75% LiDAR model.

## 5.1 STOPPING AT A TRAFFIC SIGNAL

Traffic Signals are used to impose an ordering on traffic flow direction in busy 4-way intersections. We simulate a 4-way intersection, where agents need to reach the opposite road pocket in minimum time. We constrain the agents to move along a straight line path. We employ the fixed track model and agents learn to control their accelerations to reach their destination. We study the agent behaviors by varying the number of training agents and their perception noise (Figure 2 & 3).

Note that the agents merely observe a ternary value representing the traffic light's state, not color. To make the plots in this section, we visually inspect rollouts for each converged policy to find a permutation of the ternary states that align with human red/yellow/green traffic light conventions.

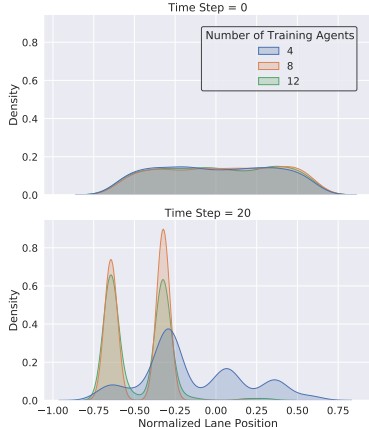

**Figure 5. Lanes emerge with more agents** With a low spatial agent density, the subpolicies converge to a roundabout motion. On increasing the spatial density, the agents learn to jointly obey the traffic signals and lanes. Agents trained with higher spatial density follow two lanes exclusively on one side of the road.

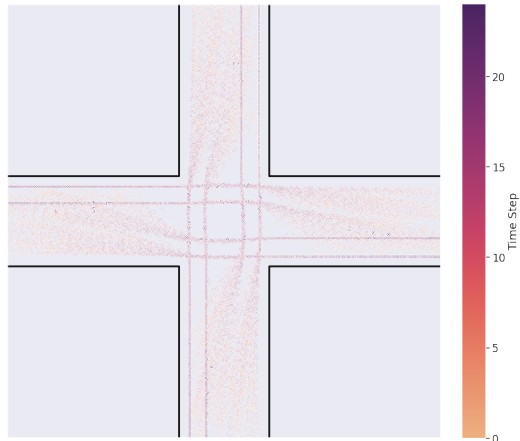

**Figure 6. Spatial Positions on Intersection Environment** Agents trained in an 8 agent environment cross the intersection in two discrete lanes on the right hand side of the road. The chosen lane depends on the starting positions of the agents. Agents starting towards the left tend to take the inner lane to allow a faster traffic flow.

## 5.2 EMERGENCE OF LANES

To analyze lane emergence, we relax the constraint on the fixed agent paths in the setup of Section 5.1. We use the Bilevel PPO Algorithm (Section 4.4) to train the two subpolicies. The spline subpolicy predicts a deviation from a path along the roads' central axis connecting the start position to the destination, similar to the GPS navigation maps used by human drivers. The acceleration subpolicy uses the same formulation as Section 5.1.

To empirically analyze the emergence of lanes, we plot the "Normalized Lane Position" of the agents over time. "Normalized Lane Position" is the directional deviation of agent from the road axis. We consider the right side of the road (in the ego frame) to have a positive "Normalized Lane Position".

Figures 4 & 5 show the variation with lidar noise and number of training agents respectively. Figure 6 shows the spatial positions of the agents for the 8 agent perfect perception environment.

## 5.3 RIGHT OF WAY

For tasks that can be performed simultaneously and take approximately equal time for completion, First In First Out (FIFO) scheduling strategy minimizes the average waiting time. In the context of driving through an intersection where each new agent symbolizes a new task, the agent that arrives first at the intersection should also be able to leave the intersection first. In other words, given any two vehicles, the vehicle arriving at the intersection first has the "right of way" over the other vehicle.

Let the time at which agent $i \in [n]$ arrives at the intersection be $(t_a)_i$ and leaves the intersection be $(t_d)_i$. If $\exists j \in [n] \setminus \{i\}$, such that $(t_a)_i < (t_a)_j$ and $(t_d)_i > (t_d)_j$, we say that agent $i$ doesn't respect $j$'s right of way. We evaluate this metric on a model trained on a nuScenes intersection (Figure 8). We observe that, at convergence, the agents follow this right of way $85.25 \pm 8.9\%$ of time (Figure 7).

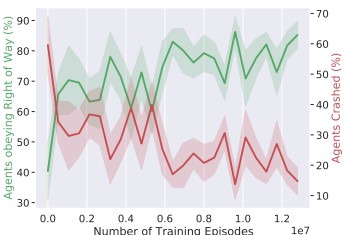

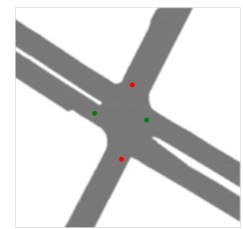

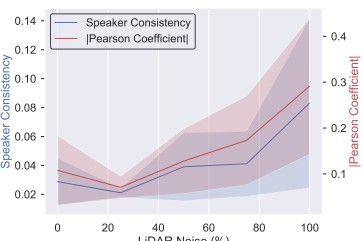

**Figure 7. Right of Way** Agents are increasing able to successfully reach their destinations (denoted by the decreasing red line) with more training episodes. The agents also increasing obey the right of way (denoted by the increasing green line). The error bars are constructed using $\mu \pm \frac{\sigma}{2}$.

**Figure 8. nuScenes Intersection** used for Right of Way (see Section 5.3) and Communication (see Section 5.4) Training and Evaluation. The red and green dots mark the location of the traffic signals and their current ternary state.

**Figure 9. Emergent Communication with Perception Noise** Speaker Consistency & Pearson coefficient between the agent's heading and its sent message increase with increase in Perception Noise. Since all agents have faulty sensors, the agents aid each other to navigate by propagating their heading to their trailing car.

## 5.4 COMMUNICATION

One way to safely traverse an intersection is to signal one's intention to nearby vehicles. We analyze the impact of perception noise on emergent communication at an intersection (Figure 8). In particular, we measure the Speaker Consistency (SC), proposed in Jaques et al. (2019). SC can be considered as the mutual information between an agent's message and its future action. We report the mutual information and the Pearson coefficient (Freedman et al., 2007) between the agent's heading and its sent message. We limit the communication channel to one bit for simplicity. Each car only receives a signal from the car in the front within $-30°$ and $30°$. Figure 9 shows that agents rely more heavily on communication at intersections when perception becomes less reliable.

## 5.5 FAST LANES ON A HIGHWAY

Highways have dedicated fast lanes to allow a smooth flow of traffic. In this experiment, we empirically show that autonomous vehicles exhibit a similar behavior of forming fast lanes while moving on a highway when trained to maximize traffic flow. We consider a straight road with a uni-directional flow of traffic. Agents are spawned at random positions along the road's axis.

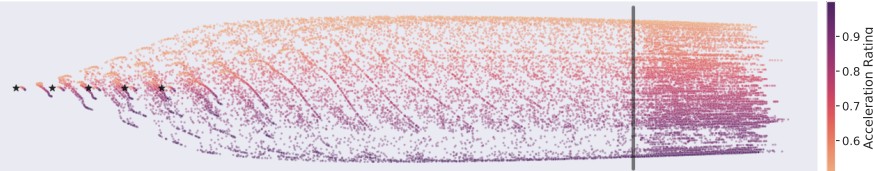

**Figure 10. Spatial Positioning in Highway Environment** Agents start from one of the positions marked by ⋆, and need to reach the end position represented by the solid black line. The agents with a higher acceleration rating move on the right hand side while the lower rated ones drive on the left hand side.

Every agent is assigned a scalar value called "Acceleration Rating," which scales the agent's acceleration and velocity limits. Thus, a higher acceleration rating implies a faster car. The spline subpolicy predicts an optimal spline considering this acceleration rating. Even though the agents can decide to move straight by design, it is clearly not an optimal choice as slower cars in front will hinder smooth traffic flow. Figure 10 shows that agents are segregated into different lanes based on their Acceleration Rating. Figure 11 visualizes this behavior over time.

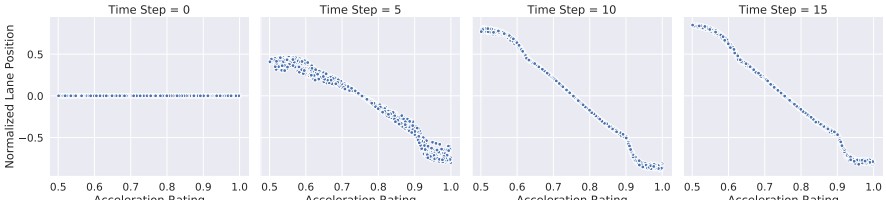

**Figure 11. Fast Lane Emergence** Visualization of rollouts from a 10-agent highway environment. In y-axis, we show the agent's position relative to the axis of the road normalized by road width. The agents reach a consensus where the faster agents end up on the right-hand side lane. This pattern ensures that slower vehicles do not obstruct faster vehicles once the traffic flow has reached a steady state.

## 5.6 Minimum Distance Between Vehicles

In this task, we evaluate the extent to which the agents learn to respect a minimum safety distance between agents while driving. When agents are too close, they are at a greater risk of colliding; when agents are too far, they are not as efficient traveling $a \rightarrow b$.

To derive a human-like "safety distance", we assume agents can change their velocity according to $v^2 = u^2 + 2ad$, where $v$ and $u$ are the final and initial velocities respectively, $a$ is acceleration, and $d$ is the distance for which the acceleration remains constant. Hence, for an agent to stop entirely from a state with velocity $v_0$, it needs at least a distance of $\frac{v_0^2}{2a_{max}}$ in front of it, where $a_{max}$ is the maximum possible deceleration of the agent.

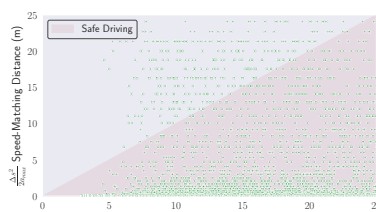

**Figure 12. Safety Distance Maintained in Nuscenes Environment** Agents need to maintain a minimum of the Speed-Matching Distance to be able to safely stop. 98.45% of the agents in this plot learn to respect this distance threshold.

Our agents perceive the environment through LiDAR; thus, agents can estimate nearby agents' velocity and acceleration. We define the safe distance as the distance needed for a trailing agent to have a zero velocity in the leading agent's frame. We assume that the leading agent travels with a constant velocity, and as such, the safe distance is defined by $\frac{\Delta s^2}{2a_{max}}$, where $\Delta s$ is the relative velocity. Any car having a distance greater than this can safely slow down. Agents trained on nuScenes Intersections obey this safety distance around 98.45% time (Figure 12).

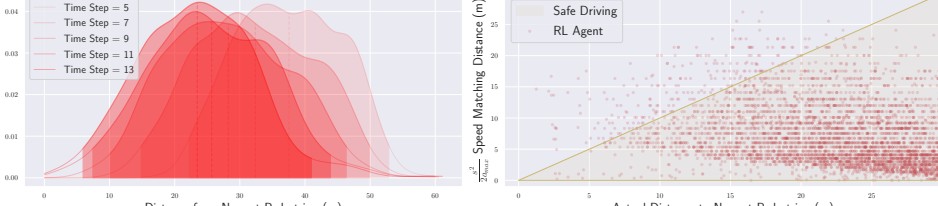

**Figure 13. Safety Distance from Pedestrians** We observe that most of the agents maintain a distance greater than the recommended speed-matching distance from the pedestrians. The shaded region in the KDE plots indicate the 95% confidence interval and the dotted line is the sample mean

## 5.7 Slowing down near a Crosswalk

In this task, we evaluate if agents can detect pedestrians and slow down in their presence. We augment the environment setup of Sec. 5.5, to include a crosswalk where at most 10 pedestrians are spawned at the start of every rollout. The pedestrians cross the road with a constant velocity. If any agent collides with a pedestrian, they get a collision reward of -1, and the simulation for that agent stops.

The KDE plots in Figure 13 show that the agents indeed detect the pedestrians, and most of them maintain a distance greater than 6m. To determine if the agents can safely stop and prevent collision with the pedestrians, we calculate a safe stopping distance of $\frac{s^2}{2a_{max}}$, where $s$ is the velocity of the agent. In the scatter plot, we observe that most agents adhere to this minimum distance and drive at a distance, which lies in the safe driving region.

## 6 DISCUSSION

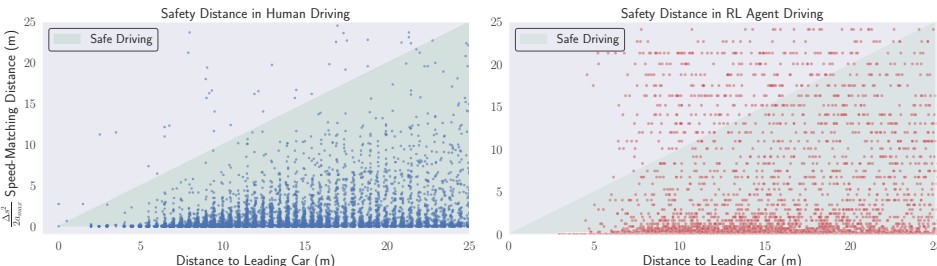

**Figure 14. Safety Distance for Humans vs. Safety Distance for RL Agents** The agents trained in our MDP (right) tend to violate the safety distance slightly more than human drivers (left) (98.45% vs. 99.56%), but in both cases a safety distance is observed the vast majority of the time (green triangular region).

### 6.1 STATISTICS OF HUMAN DRIVING

In some cases, the same statistics accumulated over agent trajectories that we use in Section 5 to quantitatively demonstrate emergence can also be accumulated over the human driving trajectories labeled in the nuScenes dataset. In Figure 14, we visualize safety distance statistics across nuScenes trajectories and safety distance statistics across RL agents trained on nuScenes intersections side-by-side. The nuScenes trainval split contains 64386 car instance labels, each with an associated trajectory. For each location along the trajectory, we calculate the safety distance as described in Section 5.6. The same computation is performed over RL agent trajectories. The agents trained in our MDP tend to violate the safety distance more than human drivers. However, in both cases, a safety distance is observed the vast majority of the time (green triangular region).

### 6.2 FUTURE WORK

By parameterizing policies such that agents must follow the curve generated by the spline subpolicy at initialization (see Section 4.4), we prevent lane change behavior from emerging. The use of a more expressive action space should address this limitation at the cost of training time. Additionally, the fact that our reward is primarily based on agents reaching destinations means that convergence is slow on maps that are orders of magnitude larger than the vehicles' dimensions. One possible solution to training agents to navigate large maps would be to generate a curriculum of target destinations, as in Mirowski et al. (2018).

## 7 CONCLUSION

In this paper, we identify a lightweight multi-agent MDP that empirically captures the driving problem's essential features. We equip our agents with a sparse LiDAR sensor and reward agents when they reach their assigned target destination as quickly as possible without colliding with other agents in the scene. We observe that agents in this setting rely on a shared notion of lanes and traffic lights to compensate for their noisy perception. We believe that dense multi-agent interaction and perception noise are critical ingredients in designing simulators that seek to instill human-like road rules in self-driving agents.

## 8 ACKNOWLEDGEMENT

This work was supported by NSERC and DARPA's XAI program. SF acknowledges the Canada CIFAR AI Chair award at the Vector Institute. AP acknowledges support by the Vector Institute.

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
