# OpenReview forum: "Emergent Road Rules In Multi-Agent Driving Environments"
_ICLR.cc/2021/Conference — ICLR 2021 Poster_

### Official Review · AnonReviewer1 · 2020-10-25
**Useful to research in autonomous driving**

**Rating:** 7
**Confidence:** 3

**Review:**

Summary: The output of the work is an MDP model that is capable of encoding complex traffic rules including traffic signals, lanes, right of way FIFO etc. Various different traffic environments such as intersections, highways, nuScenes are considered.

The MDP that results from this work is very useful for future research. Currently, there are very few simulators that encode complex traffic rules or provide the flexibility to perform research. Therefore, the possibility of new simulators resulting from this MDP is exciting and useful to encourage and promote research in autonomous driving, especially in dense and heterogeneous environments.

I am glad that the authors have provided the source code **with extensive documentation and instructions** on how to run the code and reproduce the results.

Unless there is some major flaw with this paper (that I may have missed), there is no reason to reject this paper.

---

> ### Author Response · Authors · 2020-11-17
> **Thank you for the review**
>
> We would like to thank the reviewer for the kind words. We are happy to hear that the reviewer found our work “exciting and useful” and appreciates the value in releasing these driving environments to the public. We share the vision of the reviewer that these environments will untap new areas for research in self-driving.

---

### Official Review · AnonReviewer2 · 2020-10-27
**Review from R2**

**Rating:** 5
**Confidence:** 2

**Review:**

Summary:
This paper proposes a bilevel MARL method that can learn road rules and conventions implicitly without hard coding. It is quite interesting to see a simple and straightforward idea that is effective in this task setting. However, this paper needs to be further polished in terms of its delivery, completeness, and evaluation.

Methodology:
How is the noise introduced to the mimicked LiDAR perception results?
In Spline Model, does the trajectory shape only depends on the initial state without considering the afterward interaction?
The clarity and completeness of writing could be significantly improved.
The equations, terms, and the algorithm in Section 4 need sufficient explanations. E.g., what are vf^{target}_t, H, K_1, K_2, N? What is the complete reward list?

Experiments:
It seems the route is relatively short, and the driving scenarios used for different tasks are distinct. I was wondering whether the generalization of the proposed method and the learned rules could be validated.

No baselines have been evaluated or compared. Which experimental results are or the fixed track model? What are the differences between the two models in the results? What are the reward values during the training phase?

The captions and axis labels are difficult to read.
What do the colors mean in Fig. 2?


====
My rating has been updated after the rebuttal.

---

> ### Author Response · Authors · 2020-11-17
> **Thank you for the review**
>
> We would like to thank the reviewer for their constructive and insightful feedback. We are glad that the reviewer found our approach “simple”, “interesting”, and “effective”. We address the reviewer’s questions and comments below.
> 1. “How is the noise introduced to the mimicked LiDAR perception results?” - we apologize for omitting this detail. We drop p% of lidar rays to mimic lidar noise as a proxy for "ray-dropping" LiDAR noise identified in https://arxiv.org/abs/2006.09348 .
> 2. “In Spline Model, does the trajectory shape only depends on the initial state” - Yes, the trajectory shape only depends on the initial state. We have polished the writing in the “Policy Parameterization” section to clarify this constraint.
> 3. “It seems the route is relatively short, and the driving scenarios used for different tasks are distinct” - Our maps and routes were kept purposefully short in order to probe for the emergence of different driving rules in the most direct possible way. On larger maps, we expect the same transition dynamics and reward function to result in similar emergence, but demonstrating emergence -- which is our goal in this work -- is then less targeted. We leave experiments at larger length and time scales for later work.
> 4. “I was wondering whether the generalization of the proposed method and the learned rules could be validated” - In our updated supplementary, we have added rollouts of a policy trained on a synthetic map and evaluated on a large nuScenes map. The policies transfer well to new maps without retraining.
> 5. “No baselines have been evaluated or compared” - The purpose of baselines is to offer insight on the extent to which a given approach is working relative to other methods. In the paper “Hide and Seek” https://arxiv.org/abs/1909.07528 which has a similar style of contribution to ours, the central claim of the authors is that the reward function for the game of hide and seek causes tool usage to emerge. As a result, the authors provide “baselines” in which different “curiosity-based” reward functions are used instead to prove their claim. In our work, our contribution is to show that in multi-agent driving systems with noisy perception, road rules emerge. As a result, our experiment section systematically measures the emergence of different road rules as we vary the perception quality of agents and the spatial density of agents. For us, we therefore consider the case where agents have perfect perception or when there is only one agent to be “baselines” and show that in these cases, the agents - as expected - do not exhibit road rules. If the reviewer is not convinced that we have shown that road rules emerge in multi-agent driving environments with noisy perception because a certain ablation experiment or baseline is missing, we encourage the reviewer to let us know the experiment and we will run it.
> 6. “Which experimental results are or the fixed track model?” - we have clarified which agent variant is being evaluated in each of the experiment subsections.
> 7. “What are the differences between the two models in the results?” - we have added plots comparing the emergence of traffic light usage for the fixed-track and spline agents. In general, the behavior of these two agents at convergence is very similar. The benefit of the fixed track agents is that these agents allow us to factor out lane emergence from the other behaviors that we seek to probe.
> 8. “What are the reward values during the training phase?” - we have added plots of reward vs. optimization steps to our supplementary.

---

> > ### Comment · AnonReviewer2 · 2020-11-24
> > **Thanks for the revision**
> >
> > I appreciated the detailed revision and the added supplementary, which definitely strengthened this submission in terms of its clarity in writing and visual demonstrations. Therefore, I'm willing to increase my score by 1.
> > I understood that the purpose of this paper is not to propose a novel method -- which is fine. However, while this paper did present lots of interesting observations and empirical findings, IMHO deeper investigations and validations should be included to reach the standard of a top-tier AI paper.

---

> > > ### Author Response · Authors · 2020-11-24
> > > **Thank you**
> > >
> > > We appreciate that the reviewer took the time to examine the revisions to our paper and supplementary and that the reviewer found that these revisions “definitely strengthened” our submission. Thank you for taking our revisions into account and increasing your score.
> > >
> > > We are encouraged to hear that the reviewer agrees with us that valuable contributions to the machine learning community exist that do not propose “novel methods”; to reiterate, our main contribution is instead to show the promise of a new paradigm for end-to-end self-driving in which agents are trained in a multi-agent simulator to maximize traffic flow and discover through optimization that road rules are valuable constructs.
> > >
> > > With regards to the reviewer’s remaining criticism - “IMHO deeper investigations and validations should be included” - we would appreciate it if the reviewer could be more specific about which claims in our work the reviewer believes require “deeper investigations and validations”. In our revisions, we addressed the questions included in the reviewer's initial review regarding our LiDAR simulator, reward function, transfer to new maps, and policy parameterization. We are curious to know specifically what claims the reviewer believes remain unjustified by our experiments in our revised submission.

---

### Official Review · AnonReviewer3 · 2020-10-29
**Interesting work, hope rebuttal can shed further clarity**

**Rating:** 5
**Confidence:** 2

**Review:**

The paper proposes to learn traffic rules (e.g. traffic lights, speed limits) via multi-agent RL (MARL) from observations rather than hardcoding the rules into the algorithms. To this end, the authors claims contributions in:

-- Defining a multi-agent driving environment, where each agent has incomplete observation (noisy LiDAR), and is rewarded for reaching a destination quickly without colliding. Experiments show that road rules can be learned.

-- Ablation on the choice of the MDP, and insights that perception noise and spatial density of agents are important to successfully learn the environment.

-- Authors promise to release a suite of 2D driving environments for future MARL research in self-driving.

I must admit that I am mainly a computer vision person, and I only have limited experience with RL or MARL. However, I hope that my assessment below is still better than an educated guess. I apologize in advance if there is any obvious misunderstanding.

Strengths:

-- Novelty in problem statement and at a high level: The problem statement of learning hard traffic rules via observing logs seems new. The application of MARL to the problem is new too AFAIK. And as the authors stated, the majority of multi-agent behavior works have been using imitation learning. (Maybe consider citing [1] line of work as well, which deals with multi-agent imitation learning). The proposed PPO-based method seems a good alternative.

-- Extensive experiments: There are 7 small but specific tasks such as "Traffic lights", "Emergence of lanes", etc., where the authors provided evidences to the claims that perception noise and spatial density of agents are crucial for the method's success.

[1]: Multi-Agent Generative Adversarial Imitation Learning

Weaknesses:

-- No comparison to prior work: The authors acknowledged imitation learning-based (IL) methods but did any quantitative comparison for them. I don't see any evidence why solving the proposed problem using MARL is better than IL. So why are we doing MARL over IL?

- Usefulness: for the traffic light use case, the handling of red / green lights is essentially learned by NOT driving into other cars, which obey the traffic rules. With other words, if there are no other agents to demonstrate how to behave, the agent will always prefer to run over red. This raises the question of the usefulness of the system.

-- Lack of novelty in the method itself: The paper seems to be using an off-the-shelf PPO. The centralized critic, single-step PPO, and bi-level PPO do not provide sufficient novelty in terms of methods. The centralized critic can be seen as a straightforward extension of PPO for the self-driving application. The bi-level extension is a straightforward way to optimize two objectives at the same time, which is quite common even in the ages of convex optimization (https://en.wikipedia.org/wiki/Biconvex_optimization). I think that there __is__ novelty in the method, but I am not sure if it's enough for this venue.

- Dataset: NuScenes dataset is a perception dataset, and probably does not contain many interesting interactive scenarios. Still, the method does not seem to perform very well in the experiments. E.g., according to Fig 2., many cars are still accelerating despite the red light...

Details:

- All axes names and titles in all diagrams are way too small. No way people can decipher them if printed on paper.

- No legends in the diagrams. Barely any caption. Please make diagrams self-contained if possible.

Conclusion:

This works provides an interesting new problem statement and explores the area of using MARL for autonomous driving. My main concerns regarding this paper are the lack of comparison to prior works in the experiments, and the work provides any usefulness and improvements over current autonomous systems, which are mostly based on imitation learning.

---

> ### Author Response · Authors · 2020-11-17
> **Thank you for the review (2/2)**
>
> 6. “NuScenes dataset is a perception dataset, and probably does not contain many interesting interactive scenarios” - We are not using any information from the annotated scenarios that are part of the nuScenes dataset. We *exclusively* use crops from the HD maps that come packaged with nuScenes in order to test our model on intersections with variable real-world geometry. Images of these maps can be found in the “Rollouts on Nuscenes” portion of our supplementary material. We emphasize that *all* agents in our environments use policies that are trained to minimize the time it takes to reach randomly sampled destinations without colliding; there are no fixed-trajectory agents taken from nuScenes annotations in any of our environments.
> 7. “according to Fig 2., many cars are still accelerating despite the red light” - As noted in our response to AnonReviewer4, our metrics for measuring the emergence of the different road rules in this paper are only heuristics for the qualitative behavior we seek to evaluate; there are certainly scenarios where it is appropriate to accelerate when a red light is in sight despite deceleration being the correct policy the vast majority of the time. We have added Figure 3 to further visualize that the agents respect the traffic signals and that they increasingly do so when they train in environments with many other agents and noisy perception. The reviewer may also consult videos provided in our supplementary to judge qualitatively the extent to which the agents have learned a policy that follows traffic lights.
> 8. “Details” - our captions and figures have been updated to a more appropriate font size.
> 9. “My main concerns regarding this paper are the lack of comparison to prior works in the experiments…” - Our discovery in this paper is that in a simple multi-agent driving environment, road rules found in human transportation systems emerge. Since our goal is to demonstrate emergence, the bulk of our experiments, ablations, and comparisons were designed to analyze which design choices led to this emergence. If there are specific claims that the reviewer feels were not substantiated by the ablations presented in our paper, we encourage the reviewer to recommend further ablations that we have missed.
> 10. “...and [whether or not] the work provides any usefulness and improvements over current autonomous systems, which are mostly based on imitation learning.” - To demonstrate the value of having a model of human driving behavior instead of relying on imitation learning, consider the case where construction or new traffic signs are added to a road network. Imitation learning requires re-collecting expert driving data in order to adapt to the change. In contrast, our agents can update policies in simulation by re-training in the updated environment. It is tremendously beneficial to be able to be able to model how changes to an environment will affect human behavior before taking actions in the real world.
> 11. “...which are mostly based on imitation learning.” for many of the reasons included in https://www.ri.cmu.edu/pub_files/2015/3/InvitationToImitation_3_1415.pdf, imitation learning has not been a successful approach for learning deployable driving policies; contemporary planners for self-driving are currently rule-based with a small amount of online optimization.

---

> ### Author Response · Authors · 2020-11-17
> **Thank you for the review (1/2)**
>
> We would like to thank the reviewer for their honest review and questions. We are happy to hear that the reviewer appreciates the novelty of our proposed framework for learning road rules and found our “extensive experiments” compelling. We hope that our updated submission and rebuttal will clarify the reviewer’s understanding of our work.
> 1. “The problem statement of learning hard traffic rules via observing logs seems new” - we would like to clarify for the reviewer that our approach does not use any “logs” of human driving. In contrast to the standard “behavior cloning” or “imitation learning” approaches for learning end-to-end driving policies, we are proposing to train agents to drive to randomly sampled destinations as fast as possible without colliding. We show that agents trained in this setting exhibit behavior similar to driving behavior commonly observed in human driving systems. We then analyze what features of our environment lead to the emergence of these behaviors. At no point do the agents condition on or “observe” human driving.
> 2. “So why are we doing MARL over IL?” There are numerous issues with IL as a paradigm for driving (https://www.ri.cmu.edu/pub_files/2015/3/InvitationToImitation_3_1415.pdf) including error accumulation and train/test distributional shift. RL has largely been avoided in end-to-end self-driving policies solely because it is unclear what “reward function” should be optimized and how non-player characters (NPCs) should be controlled. RL as it applies to driving has therefore been limited to reward functions that rely on human monitoring such as “distance traveled until human disengagement” (https://arxiv.org/pdf/1807.00412.pdf). In our work, we address both the reward function problem and the NPC problem by framing driving as a partially-observable MARL environment in which all agents learn to travel from $a\rightarrow b$ without colliding. Our results suggest that learning human-like driving policies in simulation is a feasible and promising direction for end-to-end self-driving.
> 3. “the handling of red / green lights is essentially learned by NOT driving into other cars, which obey the traffic rules” - We hope we can clear up this misunderstanding. As we state in the Problem Setting section, “In all our experiments, the agents share a common policy network”; there are no NPCs that have been hard-coded to follow any road rules. The agents collectively learn to follow the traffic lights because that is the optimal way for agents with noisy perception to reliably cross intersection without collisions.
> 4. “if there are no other agents to demonstrate how to behave, the agent will always prefer to run over red” - We hope we can clarify this misunderstanding. If there was only one agent, we agree that the optimal strategy for the agent to reach any destination that it samples is to ignore all traffic lights (we verify this fact in Figure 3). However, in the MARL setting, the agents all seek to reach their destination without colliding with each other and if agents receive only noisy perception about the location of other agents, it becomes optimal for the agents to leverage discrete traffic light signals for determining when it is safe to cross an intersection. This intuition is demonstrated empirically in the “traffic light usage” subsection of our experiment section. It is true that whether the agents decide to designate “green” or “red” as the signal that lets them cross the intersection is arbitrary. However, an optimal strategy must choose one convention or the other.
> 5. “Lack of novelty in the method itself” - We do not claim any novelty in the RL algorithms that we use to optimize our policies. To reiterate, our main contribution is to show that in multi-agent systems in which agents optimize for how quickly they can reach randomly sampled targets without colliding with each other, the optimal policies exhibit road rules such as lane following, traffic light usage, and fast-lane designation commonly found in human driving systems. Similar to “Hide and Seek” which also uses PPO out-of-the-box, https://openai.com/blog/emergent-tool-use/, we do not claim to introduce new RL algorithms in this paper. While end-to-end driving and trajectory forecasting methods have focused on extracting road rules from observations of expert human driving trajectories, our results demonstrate the promise of a new paradigm for end-to-end driving in which agents are trained in a multi-agent simulator to maximize traffic flow and discover through optimization that road rules are valuable constructs.

---

### Official Review · AnonReviewer4 · 2020-10-29
**Solid paper for potential accept**

**Rating:** 6
**Confidence:** 3

**Review:**

This paper investigate how to design simulation environments so the the agent trained with them can master social rules.
Cons:
1. The paper is well written and easy to read and understand. Thanks!
2. The experiments are solid and well defined.

My major concern of this paper are:
1. The authors seems to only considered the noise of sensors and the number of agents, and these two factors happened to induce social behaviors like following lanes and stopping at traffic signals. In other words, I am not quite convinced that with all vehicles being automated, sensor noise will cause them to formulate rules that what human drivers follow today. Since human driving interactions are complex, I do not think that sensor noise would be enough to induce them.
2. I would be happy to see how this configuration of simulation environment compares with reward guided social behaviors. For example, we can design a reward to encourage agents follow right of way. I think this way might be more direct and powerful.
2. The number of agents seems to be too small and may affect the formulation of road social rules.
3. It seems like the agents did not take traffic signal status as input for the action selection, then how did they formulate the signal control rules.
4. Figures 2 and 3 needs better explanations for readers to understand.

Overall, I think this paper needs better formulation of the logics and also a deep investigation of how the simulation variations lead to those behaviors.

---

> ### Author Response · Authors · 2020-11-17
> **Thank you for the review**
>
> We thank the reviewer for their valuable feedback. We are happy to hear that the reviewer found our paper “well-written” and our experiments “solid and well defined”. As mentioned in our comment to all reviewers, we would like to emphasize that while prior work has trained agents to maximize similarity to observed human driving trajectories (e.g. imitation learning), our contribution is to show that when we optimize for traffic flow, we find that agents exhibit many behaviors that have generally been assumed to require human demonstrations to learn. We address the reviewer's comments below.
>
> 1. “Since human driving interactions are complex...” - We agree with the reviewer that certain human driving behavior cannot be modeled purely as emergent phenomena in a partially-observable multi-agent system in which all users seek to go from $a\rightarrow b$ as fast as possible without colliding. For instance, we would never expect parades or funeral processions to emerge in our environments. However, our goal in this paper was to show that standard human driving behavior such as lane following or traffic light usage are optimal under the simple reward function of minimizing the time it takes to reach randomly sampled destinations without colliding with other agents. At larger length and time scales, we expect the reward function and transition dynamics used in this paper to result in other phenomena such as the emergence of lane changes, protected lefts, and one-way roads (which civil engineers have also determined maximize traffic flow in settings like New York City, for instance). Potentially, our framework could also be used to discover new driving behavior that could be adopted by human drivers to improve safety and efficiency.
> 2. “We can design a reward to encourage agents follow the right of way” - Designing complicated reward functions is what we seek to avoid in this work. Our goal is to show that the simple reward function of minimizing the time it takes agents to reach randomly sampled destinations results in much of the standard driving behavior observed in human driving systems. To clarify the purpose of the metrics plotted in our Experiment section, we use these metrics to quantitatively evaluate the extent to which the behavior of the agents trained in our MDP mimic the behavior of human drivers - not as metrics that we explicitly hope to optimize. In reality, humans themselves are unlikely to perfectly optimize for these metrics as we show in the “Discussion” section in which we use human driving trajectories in nuScenes to show that humans occasionally drive dangerously close to cars in front of them. Similarly, in “Hide and Seek” https://arxiv.org/pdf/1909.07528.pdf , “Object Movement” and “Agent Movement” are used as heuristics for tracking emergence of different hide and seek strategies.
> 3. “The number of agents seems to be too small” - Our ablation on the number of agents in the “Experiments” section shows that the number of agents used (12 per intersection in most cases) is enough to cause the emergence of all phenomena that we sought to analyze in this work. We do agree that for more complicated behaviors to emerge such as lane changes or protected lefts, it will be important to scale both the maps used in our experiments and the number of agents.
> 4. “It seems like the agents did not take traffic signal status as input” - Our agents perceive the traffic lights. We have updated our “Problem Setting” section to include details of our traffic light model.
> 5. “Figures 2 and 3 need better explanations” - We have updated the submission accordingly.
> 6. “I think this paper needs better formulation of the logics and also a deep investigation of how the simulation variations lead to those behaviors.” - The vast majority of implicit and explicit road rules that humans have developed over the last century are the result of the hard work of civil engineers seeking to define driving behavior and infrastructure that will maximize traffic flow while minimizing collisions. We find it both intuitive and exciting that in a multi-agent partially-observable driving environment, RL agents trained to optimize the same objective discover similar human-like behavior. In our experiment section, we show that the multi-agent and partially-observable nature of the driving environment are the crucial factors that lead to these behaviors. If there are specific claims in our paper that the reviewer believes we have not fully justified by our experiments, we will be happy to oblige.

---

### Author Response · Authors · 2020-11-17
**Thank you to all reviewers for their feedback**

We thank all of the reviewers for their praise and constructive feedback. We have polished the figure captions, legends, and axes as requested by AnonReviewer2, AnonReviewer3, and AnonReviewer4. A list of other edits to our paper can be found below.
* We have clarified in our abstract and introduction that the goal of our paper is to present results that suggest that, to a large extent, human driving behavior can be explained as the result of optimizing for traffic flow in a partially-observable multi-agent environment; in all of our experiments, the agents are exclusively optimize for reaching target destinations as quickly as possible without collisions. All phenomena that we analyze are “emergent” in the optimal solution to this difficult optimization problem.
*We have further emphasized that, as noted by AnonReviewer1, code and documentation for the multi-agent driving environment that we use in this paper -- along with the synthetic and real-world map data that we use for intersections -- can be found in our supplementary material. Upon public release, we expect these multi-agent environments to inspire the MARL community to participate more actively in self-driving research.
*We have provided additional details, as pointed out by AnonReviewer2, about the exact reward function used, state space of the PoMDP, and the noise added to LiDAR.
* We have provided details to clarify the notation used in Section 4 and Algorithm 1.
* We have clarified which experiments use the Fixed Track Model and which ones use the Spline Model.
* We have added what percent of the time human trajectories found in nuScenes follow the safety distance (99.56%) and our RL agents follow the safety distance (98.45%).
* We have updated some of the figures to provide a better understanding of the underlying environments used and the final converged solution for the experiments. These are summarized below:

    - Figure 3 has been changed to a 2D spatial histogram to highlight the increased importance of traffic signals for environments with high spatial agent density and noisy perception.
    - Figure 6 has been added to show the spatial locations of the agents over time. It demonstrates the emergence of lanes more clearly than our previous figure.
    - Error Bars have been added to Figures 7 and 9.
    - Figure 10 has been added to show the spatial locations of the agents for the "highway" experiment.

We hope the reviewers will engage with us in further discussion of their comments and questions during the remainder of the rebuttal period.

---

### Decision · Program_Chairs · 2021-01-07
**Final Decision**

**Decision:**

Accept (Poster)

**Comment:**

This paper shows how "road rules" (e.g., implicit designation of fast lanes on a highway) naturally emerge in a multi-agent MDP. The paper shows that interesting traffic rules do emerge, and it presents a detailed analysis of the factors that lead to this emergence. The paper is complemented by documented source code, with the aim to encourage the community to further work on the topic.

The reviewers agreed that this is original work, and appreciated its simplicity. Two concerns that were recurrently voiced were that 1) there is no algorithmic innovation and 2) there is no comparison to baseline models, or more generally a better placement in the context of existing literature.

The authors provided a detailed and, to my eyes, convincing response. With respect to the two concerns above, I would go as far as saying that 1) (no algorithmic innovation) is a feature, not a bug. The paper is interesting exactly because it studies emergent phenomena after framing multi-agent driving as a standard RL problem. Concerning 2) (lack of baselines), it seems to me somewhat besides the point: The paper is not claiming state of the art on some benchmark for a new algorithm, but studying how certain implicit rules emerge in a given setup. In this sense, as the authors point out, rather than looking at alternative baselines, it is informative to look at which aspects of the setup contribute to rule emergence, which is what the paper does.

Although I realize that in proposing this I am going beyond the reviewers' ratings, I found this to be an original and exciting paper, that I would strongly like to see accepted at the conference.